# Screening anticancer activity by Brine shrimp lethality test of extracts of *Annona stenophylla* (Engl. & Diels), *Strophanthus petersianus* (Klotzsch) and *Synadenium glaucescens* (Pax)

Roberto Luis Nhamussua[ID][1*¤], Faith Philemone Mabiki[2], Alinanuswe Joel Mwakalesi[2], Lyndy Joy McGaw[3]

1 Department of Natural Sciences, Faculty of Natural and Exact Sciences, Save University, Massinga, Inhambane, Mozambique, 2 Department of Chemistry and Physics, College of Natural and Applied Sciences, Sokoine University of Agriculture, Morogoro, Tanzania, 3 Phytomedicine Programme, Department of Paraclinical Sciences, Faculty of Veterinary Science, University of Pretoria, Pretoria, South Africa

¤ Current Address: Department of Chemistry and Physics, College of Natural and Applied Sciences, Sokoine University of Agriculture, Morogoro, Tanzania
* rluisnha@gmail.com

## Abstract

Cancer continues to be one of the main public health challenges, driving the search for new compounds with therapeutic potential. Medicinal plants represent a valuable promising source of bioactive metabolites, and the Brine Shrimp Lethality Test has been widely used as a preliminary tool to assess the toxicity of natural extracts, providing clues to their possible anticancer activity. In this study, the cytotoxicity of the extracts of *Annona stenophylla* (Engl. & Diels), *Strophanthus petersianus* (Klotzsch), and *Synadenium glaucescens* (Pax) was investigated using the BSLT as a first step in screening for potential anticancer compounds. The plant materials were harvested in Tanzania and air-dried in the shade, and ground. The extracts were prepared by total sequential solvent extraction using cold maceration, starting with ethyl acetate, followed by methanol. A total of 24 ethyl acetate and methanolic extracts were obtained from the leaves, stem bark, stem wood, root wood and root bark of the three plants studied. The toxicity of the extracts was assessed by exposing *Artemia salina* nauplii to different concentrations of the extracts, with mortality recorded after 24 h. The $LC_{50}$ was determined to evaluate the toxicity of each extract. All the extracts from the three plants exhibited different degrees of toxicity, with *A. stenophylla* demonstrating the lowest $LC_{50}$ values, indicating the highest toxicity. The methanolic extract of *A. stenophylla*'s root wood exhibited the highest toxicity, producing a mortality rate of 99.44%, corresponding to an $LC_{50} < 20$ µg/mL. The observed toxicity suggests the presence of bioactive compounds with potential anticancer activities. The results support the potential of *A. stenophylla*, *S. petersianus* and *S. glaucescens* as sources of bioactive compounds with possible anticancer activity. Further studies, including

**Data availability statement:** All data supporting this study are contained in the manuscript and Supporting Information files.

**Funding:** This study was supported by the Regional Scholarship and Innovation Fund/ Partnership for Skills in Applied Sciences, Engineering and Technology (RSFI/PASET) in the form of a grant awarded to RLN, grant N° P165581 and Save University in the form of a salary for RLN. The specific roles of this author are articulated in the 'author contributions' section. The funders had no role in study design, data collection and analysis, decision to publish, or preparation of the manuscript.

**Competing interests:** The authors state that there is no conflict of interest.

phytochemical analysis and *in vitro* anticancer assays, are recommended to identify and characterize the active constituents responsible for the observed cytotoxic effects.

## Introduction

Despite advances in technological development and cancer treatment strategies, cancer continues to be an increasingly common disease and is considered one of the main causes of death worldwide [1]. It is characterized by uncontrolled cell growth that spreads abnormally throughout the body due to aberrations in numerous cell signaling circuits [2]. Different types of cancer are typically named according to the type of tissue or organ in which they originate [3]. The causes of most cancers are still unknown, nevertheless, factors such as lifestyle habits (e.g., smoking), excessive body weight, hormonal influences, and non-modifiable factors like genetic aberration are considered major contributors [3,4]. Almost every year, millions of new cases and deaths from cancer are recorded. According to the International Agency for Research on Cancer (IARC), in 2022, there were close to 20 million new cases of cancer and 9.7 million deaths, including non-melanoma skin cancer [5]. The same source reports that, based on projections from 2022 to 2050, the global cancer burden is expected to increase by 77%. Therefore, new strategies or compounds need to be discovered to provide effective treatment with fewer adverse effects because most cancer treatments, such as chemotherapy and radiotherapy, can cause fatigue, nausea, vomiting, hair loss, dizziness, lack of appetite and others, which in most cases, lead to the patient abandoning the treatment [6–8]. Medicinal plants have been used for years and are widely explored as a natural source of active compounds for treating cancer and other diseases. Of the total number of clinically approved anticancer drugs, it can be said that natural products contribute more than 25% [9]. For instance, vinblastine and vincristine are anticancer drugs isolated from *Catharanthus roseus* and clinically approved for cancer treatment [3].

Similarly, in the present study, anticancer screening was carried out using the Brine shrimp lethality test (BSLT) method of extracts of *Annona stenophylla*, *Strophanthus petersianus* and *Synadenium glaucescens* as the basis for the development of the species under study as raw material herbal medicine.

The plants were selected based on their use in folk medicine for the treatment of various diseases including cancer [10,11], and the fact that members of their families are well-known as sources of bioactive compounds with various pharmacological properties, including anticancer [12–14]. *A. stenophylla* belongs to the Annonaceae family, is known for its bioactive metabolites such as acetogenins, flavonoids and alkaloids with cytotoxicity and anticancer potential [15]. *S. petersianus* is a member of the Apocynaceae family and belongs to a group of plants traditionally used in African medicine to treat many diseases, including cancer. Furthermore, from *C. roseus,* another member of this family, were isolated vinblastine and vincristine, the first natural drugs used in cancer therapy and still being

the most used in cancer treatment [16]. Some species of the *Strophanthus genus* are known to contain cardiac glycosides that may exhibit cytotoxicity effects [17,18]. Due to its bioactive compounds, *S. glaucescens,* from the Euphorbiaceae family, has been used in traditional medicine for various diseases, including cancer [19]. For instance, Euphol and Lupeol are the terpenes isolated from *S. grantii* and *glaucescens* [14,20,21], and β-sitosterol, a steroid isolated from diverse species including *S. glaucescens* [14,22,23], are anticancer agents and therefore candidates for cancer treatment. Further, Euphol revealed *in vitro* cytotoxicity against B16F10 melanoma cell lines, and an *in vivo* assay showed a significant reduction in tumour volume in melanoma-bearing mice [20], while Lupeol demonstrated anti-neoplastic effects against A549, a human non-small cell lung cancer cell line [21], and β-sitosterol exhibited cytotoxic activity against MCF-7 cancer cell lines [22].

*A. stenophylla* is commonly known as the dwarf custard apple in English, and is referred to as *Mtopetope* in Swahili, particularly in Tanzania [24,25]. The species has been recorded in Tanzania, Zambia, Zimbabwe, Angola, Botswana, Mozambique, the Republic Democratic of Congo and Namibia in woodland and sandy grassy slopes at the edge of wetlands [10,25]. In Tanzania, *A. stenophylla* can be found in Western, Rukwa, Tabora and Iringa Regions [25], and its fruits are particularly appreciated by herdsmen and children, who consume them for their naturally sweet, non-alcoholic juice [10]. The plant is commonly used in folk therapy as a snake repellent, for body swelling, managing diabetes, constipation, stomach pains, chest pain, blood purification, menorrhagia, dysmenorrhea, gonorrhoea, syphilis antiemetic and muscle sprains [26].

The species *S. petersianus,* commonly known as the sand forest poison rope, has been used as a poison for arrows and by the Zulus as an amulet against evil. It is native to countries from southern Kenya to South Africa [27].

*S. glaucescens* is commonly known as the milk bush plant in English and *Mvunjakongwa* in Swahili [28]. The plant is endemic in the East African Region and occurs in Tanzania, Kenya, the Democratic Republic of Congo and Burundi [29]. In Tanzania, it is distributed across diverse regions and is traditionally used for treating wounds, skin therapy, toothache, cough, tuberculosis, sexually transmitted infections, Human Immunodeficiency Virus (HIV), gastrointestinal worms and ringworms, excessive menstruation and asthma therapy [28,30–33].

The BSLT method is the first step for testing the toxicity of an extract or compound; it is also used to determine the bioactivity of a compound from a natural product. It is widely used for the pre-screening of active compounds in plant extracts and has a spectrum of pharmacological activity [34]. It is easy to perform, simple, fast and does not require a large cost with a 95% confidence level [35]. The BSLT method uses *A. salina* larvae as experimental invertebrate animals, where the toxicity of compounds is expressed by the $LC_{50}$ value. The $LC_{50}$ value indicates the concentration of compounds that causes the death of *A. salina* larvae in 50% of the population [36]. The method was applied in this study because it has a positive correlation with cytotoxicity tests using cancer cell culture. Therefore, it is often used as a tool for screening anti-cancer compounds [37].

Previous studies of *A. stenophylla* have assessed the antioxidant activity of its extract [38]. On another hand, researchers have investigated whether the root extract can inhibit α-glucosidase and α-amylase enzyme in the presence of carbohydrate substrates, suggesting a possible mechanism of its antidiabetic activity [39]. Additional studies include acute and subacute toxicity tests of its roots using rats, as well as, screening its roots and leaves by BSLT of [26,40]. No studies have been found regarding the species *S. petersianus*. Among the various studies conducted on *S. glaucescens*, Mabiki [30], assessed the toxicity of extracts from the root bark, root wood, stem bark, stem wood and leaves using the BSLT method. The findings suggested that these extracts possess potential anticancer properties.

Despite their use in folk medicine and the information outlined above, reports on screening for anticancer activity using the BSLT remain limited. This study aims to evaluate the potential anticancer activity of these plant extracts using the BSLT. The results provide preliminary insights into their toxicity, which may justify further investigations into their potential as sources of anticancer compounds.

 

## Materials and methods

### Plant collection and processing

The species were harvested in different locations according to their availability and abundance. The *A. stenophylla* (Engl. & Diels) and *S. petersianus* (Klotzsch) species were harvested from Pugu Forest Reserve in Dar es Salaam and Msubugwe Forest Reserve in Pangani district, respectively in February 2024. *S. glaucescens* (Pax) was collected in Mtumbatu, Tanzania, in October 2023. The botanical professional identified the plant species and registered the voucher specimens deposited for reference at the Institute of Traditional Medicine Herbarium (ITMH) with the numbers and coordinates as illustrated in Table 1.

The plants under study were collected using sustainable methods, which involved harvesting only the specific parts required for the research [41]. This approach aimed to minimise waste, prevent overharvesting, and ensure the long-term preservation of plant populations. All species were accurately identified by a qualified botanist, and representative samples were conserved for future research. Plant materials were manually harvested using appropriate tools such as pruning shears and small knives, depending on the plant part. Leaves were handpicked to avoid damaging the delicate structure; stems and bark were carefully cut using pruning shears. Similarly, roots were excavated with hoes and subsequently sectioned along with their bark using machetes. Fruits were not available at the time of collection. All materials were placed in breathable paper bags to avoid moisture retention during transport.

To avoid the decomposition of temperature light-sensitive compounds, all collected plant materials were air-dried under the shade (approximately 25 °C) in the Department of Chemistry and Physics laboratory at the Sokoine University of Agriculture. The plant materials were dried until they achieved a constant weight, determined over three consecutive days of weighing. The leaves of *A. stenophylla* and *S. petersianus* required 12 days to reach this point, whereas the leaves of *S. glaucescens,* due to their higher moisture content, required 23 days to reach a constant weight. The bark and wood of *A. stenophylla* and *S. glaucescens* were dried separately, and they took approximately 15 days to dry completely. For *S. petersianus,* the roots and stem were not separated from their bark and required 15 days to dry. The dried materials were ground using an electric mill (Silver Crest brand) to ensure uniform particle size, then weighed and packaged in polyethene zipper bags.

### Ethics statement

This study did not involve human participants or vertebrate animals and therefore did not require ethical approval. Field collection of plant material was conducted with permission from the Directorate of Postgraduate Studies, Research, Technology Transfer and Consultancy (DPRTC) of Sokoine University of Agriculture under reference number SUA/DPRTC/PYT/D/2022/0001/08. The study complied with all institutional, national, and international guidelines for the collection and use the plant materials in research.

### Extraction

Each harvested plant part (Table 2) was subjected to total and sequential solvent extraction using cold maceration, starting with ethyl acetate (EtOAc) to extract low to medium polarity compounds, followed by methanol (MeOH) to obtain more polar constituents [14,19].

**Table 1. Voucher specimen numbers and coordinates of the species harvested.**

| Species | Voucher specimen | Coordinates |
|---|---|---|
| *A. stenophylla* | SH1502 | UTM 0º5'05.632" S and 9º2'33.735" E 203 m |
| *S. petersianus* | SH1501 | UTM 0º4'68.350" S and 9º3'89.255" E 109 m |
| *S. glaucescens* | SH1500 | UTM 6º8'53.334"S and 36º59'21.615" E |

**Table 2. Harvested plant parts and corresponding masses used for extraction.**

| Species/parts | Root bark | Root wood | Stems bark | Stems wood | Leaves |
|---|---|---|---|---|---|
| *A. stenophylla* | 1500 g | 2400 g | 1500 g | | 1900 g |
| *S. petersianus* | 1400 g | | 2000 g | | 1700 g |
| *S. glaucescens* | 2000 g | 1600 g | 1000 g | 1600 g | 1000 g |

For this process, approximately 300 g or 400 g of each plant part (depending on its availability and density) was packed in amber bottles and extracted sequentially with 2 L of solvent, beginning with EtOAc and followed by MeOH. The mixture in the bottle was manually shaken for around 4 minutes and stored in a dark place to continue the extraction process. The extraction process for each solvent lasted for 72 h at room temperature, and each extraction was repeated three times, thereby ensuring maximum extraction [14]. The extracts were filtered using Whatman Nº 1 filter paper under gravity-assisted filtration. The resulting filtrate was then concentrated in a rotary evaporator (BUCHI), air-dried to evaporate the remaining solvent, placed in the desiccator to remove moisture, weighed and stored at − 20 °C for further analysis. A total of 24 crude extracts (EtOAc and MeOH) were obtained, their yield were calculated using the formula provided below and the yields values can be viewed within Table 3 of the results section.

$$Yield\ (\%) = \frac{Amount\ of\ dried\ crude\ extract}{Amount\ of\ dry\ sample\ used} x100$$

## Toxicity test with brine shrimp lethality test method

The toxicity screening of all extracts was conducted using the BSLT method according to Credo et al. and Meyer et al. [19,42] with slight modifications. The brine shrimp eggs and artificial sea salt were supplied by the Laboratory of the Department of Chemistry and Physics, Sokoine University of Agriculture. The eggs were hatched in a rectangular container (22 x 32 x 6 cm) consisting of two unequal compartments connected by multiple small holes (Fig 1). Both compartments were filled with artificial seawater prepared by dissolving 3.8 g of crude artificial sea salt in 1 L of distilled water.

The crude artificial sea salt used was produced in the Laboratory by evaporating seawater collected from the Indian Ocean in Dar es Salaam, as detailed in (S1 File, Appendix 1).

Approximately 50 mg of eggs were spread in larger, darkened compartments. The smaller compartment was illuminated using a white LED lamp positioned approximately 10 cm above the water surface to stimulate phototactic movement and support larvae hatching. After 24 h, hatched *A. salina* larvae migrated toward the smaller, illuminated compartment due to their phototactic behaviour [42,43]. The larvae were collected from the lighted side, which was separated from the egg-shells by a divider, after 48 h, using a 9-inch disposable pipette for use in the bioassay. The collection time allowed for complete hatching ensured adequate, viable, uniformly developed nauplii for the toxicity assay [43,44].

## Test sample preparation and implementation

Each EtOAc and MeOH extract obtained through sequential extraction was independently evaluated under the same assay conditions using the BSLT to determine their respective $LC_{50}$ values. The bioassay was conducted according to the procedures described by Pohan et al. and Meyer et al. [34,42] with slight modifications. A total of 40 mg of each extract was accurately weighed, and dissolved in 1 mL of 1% dimethyl sulfoxide (DMSO) to prepare the stock solution. From this stock, a series of working solutions was prepared by dilution with artificial seawater to achieve final concentrations of 360, 240, 180, 80, 40, and 20 μg/mL, which were used to treat *A. salina* larvae.

**Table 3. Percentage Yield of extracts and LC$_{50}$ in BSLT *in vivo* study of extracts of *A. stenophylla*, *S. petersianus* and *S. glaucescens* vs toxicity classification according to Meyer and Clackson index.**

| Extracts tested | Extract (g) | % Yield (w/w) | Mortality values | Average number of dead larvae | Mortality (%) | LC$_{50}$ (µg/mL) | Toxicity classification |
|---|---|---|---|---|---|---|---|
| Root bark of *A. stenophylla* EtOAc | 41.09 | 2.74 | 170 | 9.44 | 94.44 | < 20 | High toxic |
| Root bark of *A. stenophylla* MeOH | 87.82 | 5.85 | 86 | 4.90 | 47.78 | 193.86 | Medium toxic |
| Root wood of *A. stenophylla* EtOAc | 60.97 | 2.54 | 175 | 9.72 | 97.22 | < 20 | High toxic |
| Root wood of *A. stenophylla* MeOH | 40.05 | 1.69 | 179 | 9.94 | 99.44 | < 20 | High toxic |
| Stems of *A. stenophylla* EtOAc | 20.28 | 1.40 | 172 | 9.55 | 95.56 | < 20 | High toxic |
| Stems of *A. stenophylla* MeOH | 90.08 | 6.00 | 172 | 9.55 | 95.56 | < 20 | High toxic |
| Leaves of *A. stenophylla* EtOAc | 235.00 | 12.42 | 148 | 8.72 | 82.22 | < 20 | High toxic |
| Leaves of *A. stenophylla* MeOH | 291.00 | 15.32 | 94 | 5.33 | 52.22 | 80.00 | High toxic |
| Roots of *S. petersianus* EtOAc | 24.64 | 1.76 | 55 | 2.83 | 30.56 | 360 | Medium toxic |
| Roots of *S. petersianus* MeOH | 68.68 | 3.43 | 32 | 1.28 | 17.78 | 360 | Medium toxic |
| Stems of *S. petersianus* EtOAc | 27.01 | 1.35 | 29 | 1.61 | 16.11 | 98.65 | High toxic |
| Stems of *S. petersianus* MeOH | 26.26 | 1.31 | 61 | 3.38 | 33.89 | 360 | Medium toxic |
| Leaves of *S. petersianus* EtOAc | 164.50 | 9.25 | 6 | 0.33 | 3.33 | >360 | – |
| Leaves of *S. petersianus* MeOH | 40.88 | 2.30 | 44 | 2.44 | 24.44 | 330.23 | Medium toxic |
| Root bark of *S. glaucescens* EtOAc | 114.00 | 5.70 | 149 | 8.27 | 82.78 | <20 | High toxic |
| Root bark of *S. glaucescens* MeOH | 114.00 | 12.00 | 101 | 5.61 | 56.11 | 47.95 | High toxic |
| Root wood of *S. glaucescens* EtOAc | 31.00 | 1.57 | 153 | 8.50 | 85.00 | <20 | High toxic |
| Root wood of *S. glaucescens* MeOH | 26.43 | 1.32 | 162 | 9.00 | 90.00 | <20 | High toxic |
| Stem bark of *S. glaucescens* EtOAc | 107.00 | 6.73 | 135 | 7.50 | 75.00 | <20 | High toxic |
| Stem bark of *S. glaucescens* MeOH | 67.91 | 4.24 | 17 | 0.94 | 9.44 | <360 | – |
| Stem wood of *S. glaucescens* EtOAc | 30.01 | 1.50 | 17 | 0.94 | 9.44 | <360 | – |
| Stem wood of *S. glaucescens* MeOH | 37.27 | 1.86 | 82 | 4.56 | 45.56 | 135.67 | Medium toxic |
| Leaves of *S. glaucescens* EtOAc | 65.30 | 6.53 | 68 | 3.78 | 37.78 | 259.80 | Medium toxic |
| Leaves of *S. glaucescens* MeOH | 54.20 | 5.40 | 54 | 3.56 | 35.56 | 300 | Medium toxic |
| Positive control (Leaves of *C. roseus* MeOH) | 0.99 | 15.03 | 32 | 1.78 | 17.8 | 360 | Medium toxic |

Ten *A. salina* larvae 2 days old, were transferred into each well of a transparent, flat-bottomed 24-well plate (untreated), followed by the addition of 3 mL per well of the respective test sample. The experimental groups included the test samples, a negative control consisting of 1% DMSO in artificial seawater and a positive control, the methanolic leaf extract of *Catharanthus roseus.* The *A. salina* larvae in the experimental group were exposed to the test samples at the specified concentration. The control group received only 1% DMSO artificial seawater, while the positive control group was tested with *C. roseus* extract, known for its cytotoxic properties [45]. The experiment was conducted under consistent environmental conditions, such as a clean, well-ventilated area maintained at approximately 25 °C under continuous ambient light (~1000 lux), to simulate natural conditions and support the normal behaviour of *A. salina* larvae during the 24 h exposure period. Each concentration was tested in triplicate (n = 3 wells per concentration), providing a total of 30 larvae per concentration.

After 24 h of exposure, the number of live and dead larvae was recorded, and the mortality rate was calculated using the following equation. Larvae were considered dead if they showed no movement during observation under a magnifying lens, even after a gentle agitation.

$$\% \ death = \frac{number \ of \ dead \ nauplii}{number \ of \ the \ nauplii \ added \ in \ a \ vial} x100$$

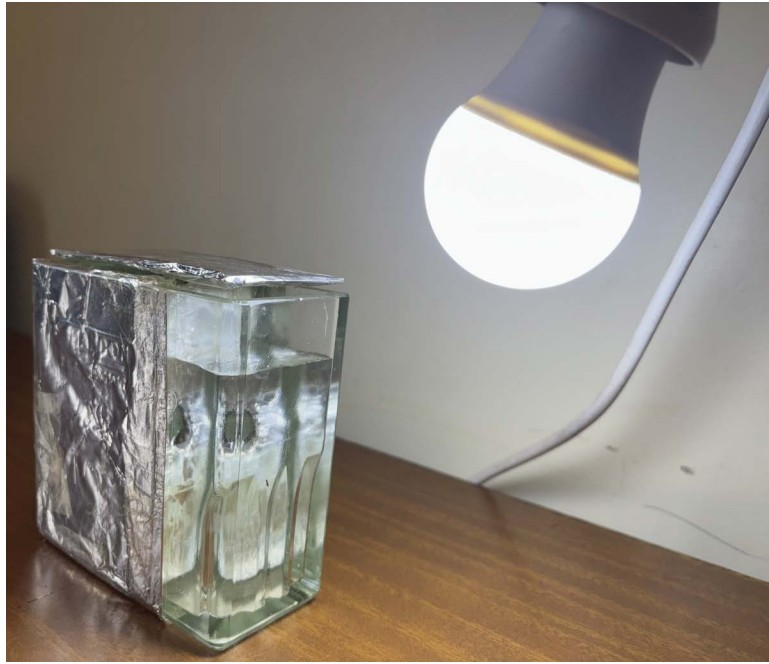

**Fig 1. Hatching container for *Artemia salina* egg incubation.**

## Statistical analysis

The median lethal concentration ($LC_{50}$) values of *A. salina* (Brine shrimp) in both control and treatment groups were estimated using the four-parameter Hill equation [46].

$$Y = Bottom + \frac{(\text{Top} - \text{Bottom})}{(1 + (X / \text{IC50})^{\text{HillSlope}}}$$

Where:
Y represents the percentage mortality
X represents concentration in µg/mL.
*Bottom* and *Top* represent the minimum and maximum response values, respectively, and $IC_{50}$ corresponds to the concentration causing 50% mortality.

Non-linear least squares regression was performed using Python's SciPy curve fit function with the Levenberg-Marquardt algorithm [47]. Model goodness-of-fit was assessed using the coefficient of determination ($R^2$), calculated as:

$$R^2 = 1 - \frac{SS_{res}}{SS_{tot}} + \dots$$

Data handling and visualization were performed in Python 3 x using NumPy, SciPy, Pandas, and Matplotlib libraries.
Toxicity classification was based on established criteria. According to Meyer's toxicity index, extracts with $LC_{50} < 1000$ µg/mL are considered toxic, while those with $LC_{50} > 1000$ µg/mL are considered non-toxic [42,48]. Clarkson's toxicity index provides further organization:
$LC_{50} > 1000$ µg/mL → non-toxic

LC$_{50}$ 500–1000 µg/mL → low toxic

LC$_{50}$ 100–500 µg/mL → moderate toxicity

LC$_{50}$ 0-100 µg/mL → high toxic [36,48].

## Results

The toxicity of plant extracts and the positive control was evaluated using the BSLT assay, and the corresponding LC$_{50}$ values are summarised in Fig 2. As illustrated, substantial variation in LC$_{50}$ values was observed among the different plant species, plant parts, and extraction solvents, reflecting distinct toxicity profiles.

The extracts of *A. stenophylla* consistently exhibited lower LC$_{50}$ values, indicating high toxicity compared to both the control and other tested extracts. In contrast, *S. petersianus* displayed LC$_{50}$ values comparable to the positive control, with the leaves EtOAc extract showing an LC$_{50}$ > 360 µg/mL. Meanwhile, the roots, (bark and wood), stems bark (EtOAc extract), stem wood MeOH extract, and leaves (EtOAc and MeOH) of *S. glaucescens* demonstrated LC$_{50}$ values which were lower than that of the positive control.

The percentage yield (w/w) of the extracts, mortality values including average, percentage of mortality of *A. salina* larvae, and toxicity classifications for all extracts tested are summarised in Table 3. The percentage yield represents the proportion of extract obtained relative to the total biomass initially subjected to extraction. Variation in extraction yield was observed across plant species, plant parts and solvents.

The mortality data revealed substantial variation in toxicity among the tested extracts. Across the concentration range of 360–20 µg/mL, LC$_{50}$ values varied from less than 20 to higher than 360 µg/mL. High lethality against *A. salina* nauplii (LC$_{50}$ < 100 µg/mL) was particularly evident in extracts derived from the roots, stems, and leaves of *A. stenophylla* in both solvents (EtOAc and MeOH), in the MeOH extract from the stems of *S. petersianus*, and the root bark and wood of *S.*

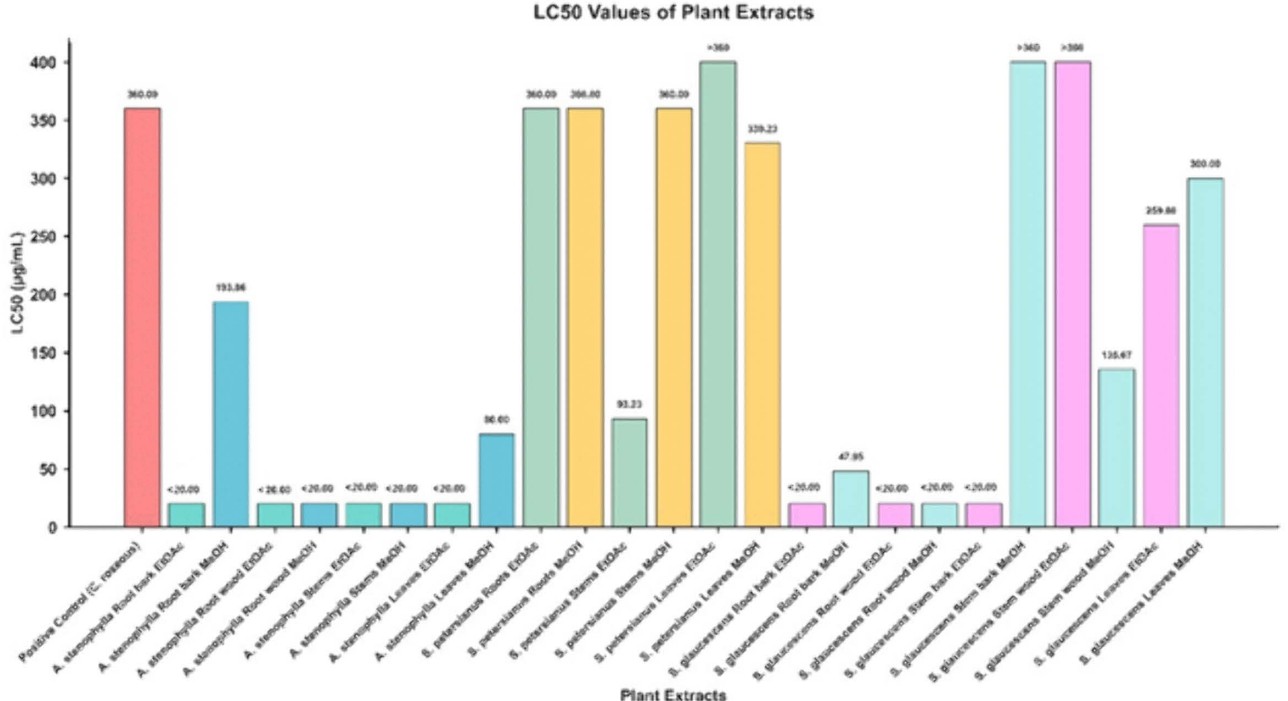

**Fig 2. LC$_{50}$ of the extracts compared with positive control.**

*glaucescens.* The MeOH extract from the root wood of *A. stenophylla,* exhibited the highest toxicity, causing a 99.4% mortality rate in the *A. salina* [42,48]. In contrast, the MeOH extract from the root bark of *A. stenophylla*; the EtOAc and MeOH extracts from the roots; the EtOAc extract from the stems; and the EtOAc and MeOH extracts from the leaves of *S. petersianus*; as well as the EtOAc and MeOH extracts from the root bark and wood, the MeOH extract from the stem bark and stem wood, and the EtOAc and MeOH extracts from the leaves of *S. glaucescens* exhibited moderate toxicity ($LC_{50}$ 100–500 µg/mL). The EtOAc extract from leaves of *S. petersianus*, MeOH extract of root bark and EtOAc extract of stem wood from *S. glaucescens* exhibited the lowest toxicity, with an $LC_{50} < 360$ µg/mL [48]. Supporting data are provided in S1 File Tables 1–26. These results suggest that both the plant part and the extraction solvent significantly influence bioactivity.

The negative control (1% DMSO in artificial seawater) showed no mortality (0%) after 24 h, confirming that the solvent and experimental conditions did not affect the viability of *A. salina* larvae. In contrast, the positive control (MeOH extract of *C. roseus*), known for its cytotoxic activity, resulted in an $LC_{50}$ of 360.00 µg/mL, thereby validating the sensitivity of the assay [49]. These control results confirm the reliability of the BSLT performed in this study.

## Discussion

Assessing the toxicity of plant extracts in cancer studies is crucial since many natural compounds have selective cytotoxic potential against tumor cells [50]. However, it is essential to determine their safety for healthy cells to develop effective therapies with minimal adverse effects [51]. This study searches for sources of potential antitumor agents through general toxicity, contributing to the development of new therapeutic agents. The BSLT method was used for screening the toxicity of the extracts from the species under study because this method has been shown to correlate positively with cytotoxic effect on cancer cells [52]. It is commonly employed as a pre-screening test to determine the mortality rate, which are directly proportional to the concentration of the extracts in drug discovery studies [53].

The results of this study are summarised in Table 3 of the results section. Thus, at the concentrations tested, the results demonstrate a wide range of toxicity among the plant extracts. The extracts of *A. stenophylla* and *S. glaucescens* exhibited relatively low $LC_{50}$ values ($LC_{50} < 100$ µg/mL), indicating strong cytotoxic activity [48]. These findings suggest the presence of potent bioactive compounds with various pharmacological properties, including potential anticancer activity [36,54]. Although specific compound confirmation was not conducted in this study, the observed cytotoxic profiles are consistent with previous reports linking such activity to secondary metabolites, phenolic compounds, alkaloids, terpenoids, acetogenins, terpenes, tannins and other secondary metabolites in related studies [55,56]. Therefore, these extracts represent priority candidates for bioassay-guided fractionation, phytochemical profiling and targeted isolation of active components. In contrast, several other extracts exhibited moderate toxicity ($LC_{50} > 100 < 500$ µg/mL), which could contain lower-abundance active constituents that require concentration or enrichment [57,58]. The EtOAc extract from leaves of *S. petersianus,* MeOH extract of stem bark and EtOAc extract of stem wood from *S. glaucescens* displayed low toxicity ($LC_{50} < 360$ µg/mL) and are unlikely to be the major source of cytotoxic principles under the extraction conditions used [59].

Among the extracts screened, the methanolic extract from the root wood of *A. stenophylla* exhibited the highest toxicity, as the lowest concentration tested (20 µg/mL) caused 96.7% mortality in *A. salina* larvae (S1 File, Table 4). The pronounced toxicity of this species may be attributed to its diverse biological activities, including antioxidants of *A. stenophylla* reported by Maroyi [10], and the presence of polyphenolic compounds such as flavonoids [38]. This is further supported by the exceptionally low $LC_{50}$ values less than 20 µg/mL observed for both the methanolic extracts root wood and root bark of *A. stenophylla* [48].

In general, at the tested concentrations (360, 240, 180, 80, 40 and 20 µg/ml), the EtOAc extracts demonstrated higher toxicity against *A. salina* larvae, as indicated by their greater average mortality rates and lower $LC_{50}$ values (Table 3). In contrast, extracts from *S. petersianus* exhibited slightly higher $LC_{50}$ values compared to those of the other two plant species.

This study incorporated both positive and negative controls to validate the cytotoxic assay. The methanolic leaf extracts of *C. roseus* (positive control) exhibited an $LC_{50}$ value of 360.00 µg/mL. Based on $LC_{50}$ values obtained in

this study, together with previous reports by Meyer et al. and Kalauni et al. [42,49], several of the tested plant extracts were classified as pharmacologically active, with activities suggestive of potential anticancer properties when compared to the positive control. The negative control (1%DMSO) exhibited no significant lethality, thereby confirming the validity of the assay and the specificity of the cytotoxic response to the plant extracts. The use of these controls ensured the reliability of the assay system and allowed for a clean distinction between true cytotoxic effects and non-specific responses to the solvent.

According to Clarkson's criteria, the tested extracts can be categorised as highly, moderately, and low toxic [48]. Furthermore, the American National Cancer Institute (NCI), considers extracts with an $LC_{50} \leq 30$ µg/mL against cancer cells as promising candidates for purification and further development as anticancer agents [60,61]. In this study, certain plants parts exhibited notable cytotoxic activity based on their $LC_{50}$ values, suggesting the presence of bioactive constituents that warrant further isolation, characterization and evaluation for potential anticancer applications [62,63]. Previous studies have also reported anticancer properties in other species belonging to the families Annonaceae, Apocynaceae and Ephorbiaceae, as well as in the *genera Annona* and *Synadenium* [15,20,64].

The bioassay with extracts of *S. petersianus* revealed high $LC_{50}$ values; therefore, its extracts are considered medium and low toxic according to the Meyer and Clackson criteria index [42,48]. This might be caused by a lower concentration of bioactive constituents, including anticancer agents [57,65]. Although scientific information on *S. petersianus* remains limited, a few studies have documented its traditional use as an arrow poison by several African communities, including Zulu, attributed to its content of potent cardiac glycosides. Additionally, it has been employed in ritualistic and protective practices aimed and averting harm [66]. However, its cytotoxic and pharmacological properties, as well as its phytochemical profile, remain largely unexplored. This study contributes to addressing this knowledge gap by demonstrating the cytotoxic effects of *S. petersianus* extracts, suggesting its potential as a promising candidate for future anticancer research. Notably, other species within the *Strophanthus genus* have been reported to exhibit antimicrobial, wound-healing, antioxidant, analgesic, and anticarcinogenic properties [67,68].

The cytotoxic effects observed in this study are consistent with those by Mabiki [30], who investigated the toxic effects of *S. glaucescens* root bark and wood, stem bark, stem wood and leaves using various solvent extracts. In that study, the dichloromethane extracts exhibited the highest toxicity against *A. salina* nauplii, followed by petroleum ether and ethanol extracts. Although different solvents were used in the present study, the observed biological activity is in agreement with Mabiki's findings [30], suggesting that key bioactive compounds are present across various plant parts and remain extractable by solvents of differing polarity.

Other studies, Yang et al., Credo et al. & Babu et al. [17,19,21], involving isolation of pure compounds from *S. glaucescens* have identified several classes of secondary metabolites, including terpenes, terpenoids, steroids and hydrolysable tannins, many of which are known to possess potential antitumor activity. The consistency between these reports and the current findings strengthens the hypothesis that *S. glaucescens* contains cytotoxic constituents of pharmacological relevance. This also highlights the need for further phytochemical characterisation and bioassay-guided fractionation of *A. stenophylla, S. petersianus,* and *S. glaucescens* to isolate and identify the active principles responsible for the observed cytotoxic effects [62].

However, some limitations remain. The relatively small sample size of the collected plant material may limit the generalizability of the findings. Additionally, although standard cytotoxicity methods were applied, future studies should incorporate mechanistic assays such as apoptosis or oxidative stress markers and include a broader set of biological models to better understand the mode of action and enhance the translational relevance of the results.

## Conclusion

This study demonstrated that the extracts of *A. stenophylla*, *S. petersianus* and *S. glaucescens* exhibit significant cytotoxic activity, supporting their traditional medicinal use and validating our hypothesis that these plants contain bioactive

 

compounds with potential anticancer properties. Among the tested extracts, the methanolic extract from the root wood of *A. stenophylla* exhibited the highest toxicity, indicating the presence of potent cytotoxic constituents extractable by this solvent.

Although the use of *C. roseus* as a positive control and 1% DMSO as a negative control ensured the reliability of the results, several limitations must be acknowledged. This includes a limited sample size of the plant materials, a limited number of test organisms per well, a lack of detailed mechanistic studies, for instance, apoptosis, oxidative stress, and the absence of *in vivo* studies for validation. These factors may constrain the broader applicability of the findings.

Nonetheless, contribute to ongoing efforts to discover new anticancer agents with fewer adverse effects, as highlighted in the introduction. The observed cytotoxic provides preliminary evidence that this species could serve as a promising source of lead compounds. Future studies should focus on bioassay-guided isolation of active constituents, detailed phytochemical profiling, mechanistic studies and evaluation in more complex biological models to further assess efficacy and safety.

In light of the increasing global burden of cancer and the urgent need for safer, more effective treatments, this study represents a step forward in the scientific exploration of traditional medicinal plants as potential contributors to cancer drug discovery.

## Supporting information

**S1 File. Appendix 1.** Detailed description of the method used to prepare artificial sea salt from seawater collected from the Indian Ocean. **Tables 1–26**. Raw data on the survival of *Artemia salina* larvae after 24 hours of exposure to different concentrations of extracts and control treatments.
(DOCX)

## Acknowledgments

The Department of Chemistry and Physics, College of Natural and Applied Sciences, Sokoine University of Agriculture, Morogoro-Tanzania, for orientation in carrying out the research and study.

## Author contributions

**Conceptualization:** Roberto Luis Nhamussua.

**Data curation:** Roberto Luis Nhamussua.

**Formal analysis:** Roberto Luis Nhamussua.

**Methodology:** Roberto Luis Nhamussua, Faith Philemone Mabiki.

**Supervision:** Faith Philemone Mabiki, Alinanuswe Joel Mwakalesi, Lyndy Joy McGaw.

**Writing – original draft:** Roberto Luis Nhamussua.

**Writing – review & editing:** Faith Philemone Mabiki, Alinanuswe Joel Mwakalesi, Lyndy Joy McGaw.

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
