## [Decision Letter · Decision Letter 0]

24 Apr 2025

Dear Dr. Nhamussua,

Thank you for submitting your manuscript to PLOS ONE. After careful consideration, we feel that it has merit but does not fully meet PLOS ONE’s publication criteria as it currently stands. Therefore, we invite you to submit a revised version of the manuscript that addresses the points raised during the review process.

We look forward to receiving your revised manuscript.

Kind regards,

Arumugam Muthuvel

Academic Editor

PLOS ONE

 [This research was supported by the Partnership for Skills in Applied Sciences, Engineering and Technology - Regional Scholarship and Innovation Fund (PASET-RSIF) scholarship 2022 grant no. P165581]. 

5. We note that your Data Availability Statement is currently as follows: [All relevant data are within the manuscript and its Supporting Information files.]

Reviewers' comments:

Reviewer's Responses to Questions

**Comments to the Author**

1. Is the manuscript technically sound, and do the data support the conclusions?

Reviewer #1: Partly

2. Has the statistical analysis been performed appropriately and rigorously?

Reviewer #1: Yes

3. Have the authors made all data underlying the findings in their manuscript fully available?

Reviewer #1: Yes

4. Is the manuscript presented in an intelligible fashion and written in standard English?

Reviewer #1: No

Reviewer #1: By using a straightforward and appropriate method, the BSLT test, the authors of this study describe the possible use of three different medicinal plants as possible future cancer treatment candidates. The methodology employed allowed for analysis of different plant parts (roots, stems, leaves) and based on levels of toxicity were compared to currently known compounds with anticancer properties. The authors stressed the importance of the need for cancer treatments that have minimal adverse effects on humans, as the rate of cancer within the human population is only expected to grow over time. The hypothesis and performed experiments are fascinating and, I believe, an interesting take on discovering compounds to use as treatments for cancer, however the manuscript itself could use vast improvements in the quality of writing, level of detail, and overall level of impact it leaves the reader with.

Introduction

Line 52-53. I assume that the authors are saying that the names of different cancers come from the different organs and tissues in which uncontrolled cell growth occurs, but with the words “it is” and “they are found” this makes the sentence awkward to read, please reword.

Line 54, please add a comma behind the word ‘unknown’ to assist with grammar and sentence structure “are still unknown, nevertheless, factors such as…”

Line 60-62, could the authors include a short description of some of the adverse effects that current cancer treatments have? Or could the impact/level of “adverse” be described here? As a reader, I like that the sentence is citing other works that describe these effects, however, to read a small blip about what those are and why they are important to this current study piques my interest more than a simple citation.

Line 64: what are the authors referring to here when they say “natural products”?

Line 76, please remove the word “which” from this sentence to improve grammar and sentence structure.

Line 77-79, please reword this sentence. The comma usage as well as general word choice and syntax make this very difficult to read and understand.

Line 84, I believe the word here should be “including” rather than “inclining”.

Line 82-85. The end of this sentence needs a re-wording as singular and plural versions of “agents” are being described. If euphol and lupeol and B-sitosterol are anticancer agents then they are “candidates” for cancer treatment, not “the candidate”. If only one is being mentioned as a prime candidate, then the sentence needs to be restructured.

Line 86, Sentence rewording suggestion to improve clarity “In Tanzania, A. stenophylla’s fruits are highly sought after by herdsmen and children as these fruits produce non-alcoholic juice”. I had to read this twice to understand what was being stated.

Lines 86-96. I really like how the authors describe each of these plants’ origins, their common/medicinal uses, where they are native to and provide each of the citations for this information, I only wish that it was consistent for each of the species. I would like the common name for all of them, not only just the sand forest poison rope, and I believe that just for structure of this paragraph, if the authors keep a consistent format for how they introduce each of these species and the corresponding information, it would flow better and be more collectively informative.

Line 97, please correct from “…the toxicity of the extract…” to “…the toxicity of an extract…”

Lines 97 to 100, these two sentences could be combined with a semi-colon as they relate closely to one another and are also both independent clauses.

Lines 107-111. This sentence is far too long, and the incorrect comma usage makes it difficult to understand. Please reword.

Line 113. A study was conducted on what that assessed toxicity of each extract mentioned? Using a citation number to replace words within a sentence is not appropriate, please keep the citation, but also describe for the reader what you are referring to and the point which you are trying to make.

General Introduction Comments:

The authors use a lot of short sentences throughout the introduction, which is not a problem in the grand scheme, but it makes some areas read in a truncated manner when they could flow better for the reader. There are also a few instances of sentences that are far too long and need to be trimmed (mentioned in my specific line comments). Moreover, it seems that some sentences have incredible grammar whereas others are lacking significantly. All in all, the introduction is difficult to read from top to bottom due to this inconsistency in syntax. I highly recommend that someone go through top to bottom and correct all grammar, syntax, proper word tense, and fix incorrect usage of prepositions. This will improve the overall quality of the writing dramatically. The topic, hypothesis, experimental investigations are fascinating, but as it reads currently it is difficult for the reader to digest.

Materials and Methods

Lines 125 and 127, as the genus and species names have already been fully introduced within the introduction section, each plant species here can be referred to in the abbreviated manner “A. stenophylla, S. petersianus, and S. glaucescens”.

Line 136. Please add a comma after the word “compounds, …”

Line 138. As I, and I assume others, am unfamiliar with the method of harvesting and drying plants, could the authors describe this process in a little more detail here? How long were the plants dried in the shade, what length of time did the weight/mass need to remain consistent to be considered “dry”, did the bark and wood take longer than the roots, stems, and leaves? How did the “harvesting” actually occur, were tools used or was it a delicate hand-picking process? Was the plant matter collected in a different manner than the bark and wood?

Line 140. I believe “Silver Crest reference” relates to the electric mill device being used to ground the dried materials, but I am not sure if that is part of the name of the device or what purpose these words have within the sentence.

Line 143-146. I believe this description of plant species, parts, and their specific masses would be better described visually with a small table rather than in sentence format. It would significantly reduce the length of the first sentence of the extraction paragraph also. Suggestion: “Harvested plant parts (Table 1) were subjected to total and sequential cold maceration starting with ethyl acetate followed by methanol as solvents”

Line 147. Was 300 g or 400 g used per 2 L of solvent? Did it depend on which plant part was being macerated? Was the solvent a 1:1 ratio of ethyl acetate and methanol? Why 2 liters of solvent per 300-400g of solute, was this an established in-house protocol for extraction?

Line 151. Please move the comma here to behind the word “times” rather than behind the word “thereby” for grammatical correctness. “…was repeated three times, thereby ensuring maximum extraction”.

Line 151-154. Was the extract passed through Watman number 1 filter paper? What method was “used” pertaining to the filter paper? Please remove the comma after the word ‘was’, it should read as “was concentrated”, then please add the word “then” before the word ‘weighed’, it should read as “…remove moisture, then weighed and stored at…” Also, why negative 4 C as a temperature? 4 degrees Celsius and negative 20 degrees Celsius are two common temperatures, but was there a reason for this specific temperature?

Line 154. What exactly is a “crude extract” following a cold maceration? What was the original plant matter that formed the 24 crude extracts (stems, roots, etc.)? How many of the 24 samples were processed from each individual species of plant?

Line 154. Please correct this sentence for grammar. A suggestion reads as follows, “24 crude extracts were obtained, their yields were calculated using the formula below, and the yield values can be viewed within Table 2 of the results section.”

Line 161. Could an image/photograph of the container used to hatch the eggs be included? It is difficult to visualize with only what is written and would benefit future readers if they intend to follow a similar method described here.

Line 163. What type of sea salt was used for the artificial seawater? Could a vendor and item/catalog number be provided here?

Line 164-165. When the container is described above, could the authors also include the lighting setup? It is unclear if this is a different container, where the lights are positioned, and why the lights were necessary for the hatching of larvae. If larvae were attracted to light at 24 hours, what purpose did collecting them at 48 hours serve?

Line 159 and 167. Please list out the publications you are citing in these places and consistently throughout the manuscript as Last Name, et al. followed by the bracketed numbered citations [XX]. For example (line 158-159), “The toxicity screening of all extracts was carried out by the BSLT method according to Credo et al. and Meyer et al. [18,37] with slight modifications”.

Line 166. “Test sample preparation and implementation”. This section seems to be missing many details, and much clarification to describe to the reader the protocol and experimental process which was performed. Please elaborate. Moreover, the authors should heavily focus on grammar as sentences like “In each well plate were added 10 larvae of A. salina 2 days old, test sample and the total volume of 3 ml was completed with artificial seawater” lower the quality of writing and make it difficult for the reader to understand the true impact of the methodology employed.

Line 170-171. What type of plates were used here (6-well, 12-well, etc. ; treated vs. untreated; clear vs. opaque,)? Please provide details. Was 3 mL the total volume of seawater the A. salina were added to? This sentence needs to be reworded with a focus on proper grammar, it is illegible as is.

Line 172. How were different concentrations of sample achieved? Where did these come from – A. salina larvae in 3 mL seawater or 5 mL extract stock solutions? This sentence being placed directly after the experimental setup of brine shrimp in seawater within welled plates does not make logical sense, what process was performed to achieve these concentrations and why are there so many different concentrations (360, 240, 180, 80, 40, 20 ug/mL)? Please provide more details.

Line 173. Please define “sufficient light”. What is the purpose of placing the experiment plate within “sufficient light” for 24 hours?

Line 174. Please provide more details about the standard solution and its composition (volumes, etc.). Was this used as a negative control? Were any dead A. salina found within the standard solution throughout the experiment? Moreover, were positive controls considered for this experiment, such as another natural product that is currently being used as a cancer treatment with an LC50 that could be compared to the three plant species in focus here?

Line 182-183. Could brief additional details be added here to describe what the probit value is for readers that are unfamiliar? How is that value calculated from the percentage of A. salina death?

Line 186. Please add in “Credo et al.” prior to the citation. Consider moving the graph to the results section rather than keeping this in the methods. Moreover, please define within the text what Y means and how that differs from the Predicted Y. Is there a linear regression for each of the samples that were listed within Table 2? It is confusing to me as a reader, as I am unfamiliar with LC50 calculations, why a linear regression is being mentioned with minimal context for how these two analyses are coupled.

Line 192-197. Thank you for describing the two different toxicity indexes as that is informative to the reader. How were these indexes used within study to interpret the results?

Methods General Comments:

To summarize my above comments collectively, please add a considerable amount of detail. Although the analysis done here was straightforward and appropriate, the limited information about reagents, volumes of stock solutions, minimal description of the different concentrations created leaving the reader wondering “how and why?” reduce the ability for this sort of study to be replicated. Chosen methodology and specifics are incredibly important for future replication of the work. Also, when citing previously published methods, please also include a brief description of the key takeaways or steps that are pertinent to the current study, the less-important details can then be viewed by the reader via the included citation.

Results

Line 202-205. The way the methods section “extraction” reads makes it seem as though the protocol utilized to create plant extracts is a combination of EtOAc with MeOH, but within the first paragraph of the results section, it seems that there were a set of extracts that were only treated with EtOAc. Can the authors provide clarification, and more detail about the process used to create extracts and describe each of the conditions assessed by the BSLT test?

Line 205-207. What were the mortality values and LC50 values for each of the plants/parts? There is no mention of a figure or a table to reference within this sentence. Coming back to this, the second paragraph starting at line 208 reads in a more detailed manner and I see references Table 2. My suggestion would be to move the last two sentences (lines 203-207) to an appropriate place in the discussion section as this text reads more as discussion than results.

Table 2. How many replicates are being averaged for each of the “average number of dead larvae”? If 10 fish are placed per well, then to get an average value with a decimal place there must be multiple values that are being averaged. Please add and describe the justification for the sample size as well as number of experiments performed within the results section.

Table 2. Why, if a concentration series of 360, 240, 180, 80, 40, and 20 ug/ml was created, were concentrations of 4000, 6000, 200, 8000, and 10000 ug/ml tested and recorded as final values in table 2? The context of where the concentration series comes from and why it was made is significantly lacking, moreover, how it directly relates to the reported results is unclear.

Table 2. Please also add some description for how the % yield/g values are being reported. It does not make sense how a concentrate of, say leaves of A. stenophylla EtOAc, would have a 12.42 % yield per gram with a (standard deviation, standard error? Please clarify) of 235 as one example.

Figure 1. I was expecting to read something within the results that relates back to the linear regression analysis explained within the methods. It struck me as odd that a linear regression graph/figure was included within the methods section titled “statistical analysis”, rather than a simple explanation that a linear regression analysis was performed that is used for LC50 calculations.

Discussion

Line 228-231. Please split this sentence up.

Line 234. How do you know these bioactive compounds are phenolic, alkaloids, etc.? Were confirmation tests done to assess these compounds or is this claim supported by other research in that the LC50 values being relatively low provide evidence for this?

Line 242-245. It is counterintuitive to state that based on results, extracts show promise for isolation and purification of extracts. I believe the authors are trying to describe that the different parts of the species, based on their LC50 values, show promise as possible cancer treatments/have anticancer properties, which is also supported by the research mentioned within line 246. Please consider the message intended for the reader, and how it relates to the specific experiments done, in addition to how this study may be expanding the field of cancer treatment in a novel way.

Line 250. If information is limited for this plant, what information is available? Please include. If there is a knowledge gap with this species of plant, where minimal information is known about it then stating that alongside the possibility of it being used for cancer treatment could shape that knowledge gap into productive future research.

Results and Discussion General Comments:

Overall, these sections could both benefit from expansion, elaboration, inclusion of more details, not only experimentally, but to round out the claims the authors are making that reflect the purpose of the study. Grammar, syntax, and punctuation also need to be addressed throughout. There are no limitations included for the methods employed, sample size of plants collected and analyzed, and seemingly minimal controls or comparisons. Please expand by providing some possible future experiments or specific research questions that this research could lead to. The discussion section also provides minimal justification for why the results of this study are consistent with previous research.

Conclusion

Please expand, this current conclusion does not adequately summarize the entire study, explain key takeaways to the reader from what was found, nor does it mention limitations and future direction. This could be expanded upon with breadth of information rather than sheer length and allow for a more satisfactory closing for whether the hypothesis was approved or rejected.

Within the introduction there is mention that more cancer treatments with minimal adverse effects are needed, due to the projected growth of cancer in the human population. I find this to be a fascinating take and the investigation into the toxicity of medicinal plants an interesting approach to address this “need” within the scientific field/cancer research field. As a suggestion, mention how the results of this study help address that research need, which is described well in the introduction, here in the conclusion so that everything throughout the manuscript comes full circle.

**Do you want your identity to be public for this peer review?** For information about this choice, including consent withdrawal, please see our Privacy Policy

Reviewer #1: No

---

## [Author Response · Author response to Decision Letter 1]

26 Oct 2025

Journal requirements

Response: We thank the editorial team for the guidance. We have revised the manuscript and ensured that it conforms to the PLOS ONE style and formatting requirements. All files have been renamed according to the PLOS ONE file-naming conventions.

Response: We thank the editorial team for this comment. The Methods section has been updated to include detailed information on the field permit. Fieldwork was conducted with permission from the Directorate of Postgraduate Studies, Research, Technology Transfer and Consultancy (DPRTC) of Sokoine University of Agriculture, under reference SUA/DPRTC/PYT/D/2022/0001/08.

No additional permits were needed beyond the one issued by the DPRTC of Sokoine University of Agriculture.

3. We note that the grant information you provided in the ‘Funding Information’ and ‘Financial Disclosure’ sections does not match.

Response: We thank the editorial team for noting this discrepancy. We have corrected the information to ensure consistency between the ‘Funding Information’ and ‘Financial Disclosure’ sections.

[This research was supported by the Partnership for Skills in Applied Sciences, Engineering and Technology - Regional Scholarship and Innovation Fund (PASET-RSIF) scholarship 2022 grant no. P165581].

If this statement is not correct, you must amend it as needed.

Response: Thank you for the feedback. We have amended the statement to accurately reflect the revised information:

This study was funded by the Regional Scholarship and Innovation Fund (RSIF) of the Partnership for Skills in Applied Sciences, Engineering and Technology (PASET) (Scholarship 2022 grant N� P165581) grant to SACIDS Africa Centre of Excellence for Infectious Diseases of Human and Animals in Southern and Eastern Africa (SACIDS-ACE) at the Sokoine University of Agriculture (SUA). Roberto Luis Nhamussua is a recipient of an RSIF-PASET doctoral scholarship at SUA.

5. We note that your Data Availability Statement is currently as follows: [All relevant data are within the manuscript and its Supporting Information files.]

Response: We thank the reviewer for the guidance. We have now included the full ethics statement in the 'Methods' section of the manuscript. The addition made in the methods section of the manuscript is on lines 165-171 and reads as follows:

Ethics statement

“This study did not involve human participants or vertebrate animals and therefore did not require ethical approval. Field collection of plant material was conducted with permission from the Directorate of Postgraduate Studies, Research, Technology Transfer and Consultancy (DPRTC) of Sokoine University of Agriculture under reference number SUA/DPRTC/PYT/D/2022/0001/08. The study complied with all institutional, national, and international guidelines for the collection and use the plant materials in research”.

Reviewer comment and author's response

Introduction

Line 52-53. I assume that the authors are saying that the names of different cancers come from the different organs and tissues in which uncontrolled cell growth occurs, but with the words “it is” and “they are found” this makes the sentence awkward to read, please reword.

Response: Thank you for pointing this out. We have revised the sentence for clarity. The revised sentence now on lines 50-51 reads as follows:

“Different types of cancer are typically named according to the type of tissue or organ in which they originate”

Line 54, please add a comma behind the word ‘unknown’ to assist with grammar and sentence structure “are still unknown, nevertheless, factors such as…”

Response: Thank you for your careful reading and helpful suggestion. We have revised the sentence on lines 52-54 to read as follows:

“The causes of most cancers are still unknown, nevertheless, factors such as lifestyle habits like smoking, excessive body weight gained, hormonal contributions, as well as non-modifiable factors like genetic aberration, were among the major causes”

Line 60-62, could the authors include a short description of some of the adverse effects that current cancer treatments have? Or could the impact/level of “adverse” be described here? As a reader, I like that the sentence is citing other works that describe these effects, however, to read a small blip about what those are and why they are important to this current study piques my interest more than a simple citation.

Response: Thank you for this insightful suggestion. We have added a brief description of common adverse effects associated with current cancer treatments to provide additional context. The revised passage on lines 58-62 now reads as follows:

“Therefore, new strategies or compounds need to be discovered to provide effective treatment with fewer adverse effects because most cancer treatments, such as chemotherapy and radiotherapy can cause fatigue, nausea, vomiting, hair loss, dizziness, lack of appetite and others, which in most cases, lead to the patient abandoning the treatment”

Line 64: What are the authors referring to here when they say “natural products”?

Response: Thank you for pointing this out. We mean by natural products bioactive compounds derived from plants, microorganisms or marine organisms. To improve clarity, we have updated the sentence on lines 62-63, now reads as follows:

“Medicinal plants have been used for years and widely explored as one of the natural sources of active compounds for treating cancer and other diseases”

Line 76, please remove the word “which” from this sentence to improve grammar and sentence structure.

Response: Thank you for your suggestion. We have revised the sentence by removing the word which to improve grammar and sentence flow. The updated sentence on lines 74-75 now reads as follows:

“A. stenophylla belongs to the Annonaceae family and is known for its bioactive metabolites such as acetogenins, flavonoids and alkaloids with cytotoxicity and anticancer potential”

Line 77-79, please reword this sentence. The comma usage as well as general word choice and syntax make this very difficult to read and understand.

Response: Thank you for highlighting this issue. We have revised it to improve readability and clarity. The new sentence on lines 76-77 now reads as follows:

“S. petersianus is a member of the Apocynaceae family and belongs to a group of plants traditionally used in African medicine to treat many diseases, including cancer.”

Line 84, I believe the word here should be “including” rather than “inclining”.

Response: Thank you for catching this error. We have made the correction accordingly and the revised sentence on lines 81-82 now reads as follows:

“Due to its bioactive compounds, S. glaucescens, from the Euphorbiaceae family, has been used in traditional medicine for various diseases, including cancer.”

Line 82-85. The end of this sentence needs a re-wording as singular and plural versions of “agents” are being described. If euphol and lupeol and B-sitosterol are anticancer agents then they are “candidates” for cancer treatment, not “the candidate”. If only one is being mentioned as a prime candidate, then the sentence needs to be restructured.

Response: Thank you for the helpful observation. We have revised the sentence to correctly reflect the plural form and improve clarity. The updated sentence on lines 82-85 now reads as follows:

”For instance, Euphol and Lupeol are the terpenes isolated from S. grantii and glaucescens and β-sitosterol, a steroid isolated from diverse species including S. glaucescens, are anticancer agents and therefore candidates for cancer treatment.”

In addition, we thought it necessary to provide some details about Euphol, Lupeol and β-sitosterol to clarify why they could be considered candidates for cancer treatment. The updated brief on lines 85-89 reads as follows:

“Further, Euphol revealed in vitro cytotoxicity against B16F10 melanoma cell lines and in vivo assay showed significant reduction in tumour volume in melanoma-bearing mice, while Lupeol demonstrated anti-neoplastic effects against A549, a human non-small cell lung cancer cell line, and β-sitosterol exhibited cytotoxic activity against MCF-7 cancer cell lines”.

Line 86, Sentence rewording suggestion to improve clarity: “In Tanzania, A. stenophylla’s fruits are highly sought after by herdsmen and children as these fruits produce non-alcoholic juice”. I had to read this twice to understand what was being stated.

Response: Thank you for pointing this out. We have revised it as follows (lines 94-96):

“In Tanzania, A. stenophylla can be found in Western, Rukwa, Tabora and Iringa Regions, and its fruits are particularly appreciated by herdsmen and children, who consume them for their naturally sweet, non-alcoholic juice”

Lines 86-96. I really like how the authors describe each of these plants’ origins, their common/medicinal uses, where they are native to and provide each of the citations for this information, I only wish that it was consistent for each of the species. I would like the common name for all of them, not only just the sand forest poison rope, and I believe that just for structure of this paragraph, if the authors keep a consistent format for how they introduce each of these species and the corresponding information, it would flow better and be more collectively informative.

Response: We thank the reviewer for this helpful and encouraging comment. We agree that the consistency in the presentation of the plant species would improve the clarity and the flow of the paragraph. We have revised the format to ensure consistency for each species. The updated format on lines 90-108 reads as follows:

“A. stenophylla is commonly known as the dwarf custard apple in English, and is referred to as Mtopetope in Swahili, particularly in Tanzania. The species has been recorded in Tanzania, Zambia, Zimbabwe, Angola, Botswana, Mozambique, the Republic Democratic of Congo and Namibia in woodland and sandy grassy slopes at the edge of wetlands. In Tanzania, A. stenophylla can be found in Western, Rukwa, Tabora and Iringa Regions [26], and its fruits are particularly appreciated by herdsmen and children, who consume them for their naturally sweet, non-alcoholic juice [11]. The plant is commonly used in folk therapy as a snake repellent, for body swelling, managing diabetes, constipation, stomach pains, chest pain, blood purification, menorrhagia, dysmenorrhea, gonorrhoea, syphilis, antiemetic and muscle sprains.

The species S. petersianus, commonly known as the sand forest poison rope, has been used as a poison for arrows and by the Zulus as an amulet against evil. It is native to countries from southern Kenya to South Africa.

S. glaucescens is commonly known as the milk bush plant in English and Mvunjakongwa in Swahili. The plant is endemic in the East African Region and occurs in Tanzania, Kenya, the Democratic Republic of Congo and Burundi. In Tanzania, it is distributed across diverse regions and is traditionally used for treating wounds, skin therapy, toothache, cough, tuberculosis, sexually transmitted infections, Human Immunodeficiency Virus (HIV), gastrointestinal worms and ringworms, excessive menstruation and asthma therapy”

Line 97, please correct from “…the toxicity of the extract…” to “…the toxicity of an extract…”

Response: We thank the reviewer for this careful observation. We have revised the phrase on line 97 as suggested, changing the toxicity of the extract to the toxicity of an extract to improve the clarity and accuracy. The revised version on line 109 reads as follows:

“The BSLT method is the first step for testing the toxicity of an extract or compound”

Lines 97 to 100, these two sentences could be combined with a semi-colon as they relate closely to one another and are also both independent clauses.

Response: We appreciate the reviewer’s suggestion concerning sentence structure in lines 97-100. We have revised these sentences by combining them with a semi-colon, as recommended to enhance the flow and cohesion. The change made in the manuscript, on lines 109-110, reads as follows:

“The BSLT method is the first step for testing the toxicity of an extract or compound; it is also used to determine the bioactivity of a compound from a natural product”.

Lines 107-111. This sentence is far too long, and the incorrect comma usage makes it difficult to understand. Please reword.

Response: We sincerely appreciate your time regarding clarity and sentence structure. We have revised the sentence to improve readability and breaking it into short sentences and correcting comma usage. The change made in the manuscript on lines 119-122 reads as follows:

“Previous studies of A. stenophylla have assessed the antioxidant activity of its extracts. On another hand, researchers have investigated whether the root extract can inhibit α-glucosidase and α-amylase enzymes in the presence of carbohydrate substrates, suggesting a possible mechanism for its antidiabetic activity. Additional studies include acute and subacute toxicity tests of its roots using rats, as well as screening its roots and leaves using the BSLT method.”

Line 113. A study was conducted on what that assessed toxicity of each extract mentioned? Using a ci

---

## [Editor Report · Decision Letter 1]

29 Oct 2025

Screening anticancer activity by Brine shrimp lethality test of extracts of Annonastenophylla (Engl. & Diels), Strophanthuspetersianus (Klotzsch) and Synadeniumglaucescens (Pax)

PONE-D-25-07676R1

Dear Dr. Nhamussua,

We’re pleased to inform you that your manuscript has been judged scientifically suitable for publication and will be formally accepted for publication once it meets all outstanding technical requirements.

Kind regards,

Arumugam Muthuvel

Academic Editor

PLOS ONE
---

## [Editor Report · Acceptance letter]

PONE-D-25-07676R1

PLOS One

Dear Dr. Nhamussua,

I'm pleased to inform you that your manuscript has been deemed suitable for publication in PLOS One. Congratulations! Your manuscript is now being handed over to our production team.

Kind regards,

on behalf of

Dr. Arumugam Muthuvel

Academic Editor

PLOS One